# Effects of Hyperoxia on the Refraction in Murine Neonatal and Adult Models

**DOI:** 10.3390/ijms20236014

**Published:** 2019-11-29

**Authors:** Kiwako Mori, Toshihide Kurihara, Xiaoyan Jiang, Shin-ichi Ikeda, Ayako Ishida, Hidemasa Torii, Kazuo Tsubota

**Affiliations:** 1Department of Ophthalmology, Keio University School of Medicine, 35 Shinanomachi, Shinjuku-ku, Tokyo 160-8582, Japan; morikiwako@gmail.com (K.M.); jiaxiangya@yahoo.co.jp (X.J.); shin-ikeda@keio.jp (S.-i.I.); ayakoishida9696@gmail.com (A.I.); htorii@2004.jukuin.keio.ac.jp (H.T.); 2Laboratory of Photobiology, Keio University School of Medicine, 35 Shinanomachi, Shinjuku-ku, Tokyo 160-8582, Japan; 3Tsubota Laboratory, Inc., Keio University Shinanomachi Campus 2-5F, 35 Shinanomachi, Shinjuku-ku, Tokyo 160-8582, Japan

**Keywords:** retinopathy of prematurity, oxygen-induced retinopathy, myopia, hyperoxia, high concentrated oxygen

## Abstract

Whether hyperoxia affects the refraction in neonatal and adult mice is unknown. The mice exposed to 85% oxygen at postnatal 8 days (P8d) for 3 days and the mice exposed to normal air were assigned to the neonatal hyperoxia and normoxia groups, respectively. The refraction, the corneal curvature radius (CR) and the axial length (AL) were measured at P30d and P47d. Postnatal 6 weeks (P6w) adult mice were divided into the adult hyperoxia and normoxia groups. These parameters were measured before oxygen exposure, after 1 and 6 weeks, and every 7 weeks. The lens elasticity was measured at P7w and P26w by enucleation. The neonatal hyperoxia group showed a significantly larger myopic change than the neonatal normoxia group (P47d −6.56 ± 5.89 D, +4.11 ± 2.02 D, *p* < 0.001), whereas the changes in AL were not significantly different (P47d, 3.31 ± 0.04 mm, 3.31 ± 0.05 mm, *p* = 0.852). The adult hyperoxia group also showed a significantly larger myopic change (P12w, −7.20 ± 4.09 D, +7.52 ± 2.54 D, *p* < 0.001). The AL did not show significant difference (P12w, 3.44 ± 0.03 mm, 3.43 ± 0.01 mm, *p* = 0.545); however, the CR in the adult hyperoxia group was significantly smaller than the adult normoxia group (P12w, 1.44 ± 0.03 mm, 1.50 ± 0.03 mm, *p* = 0.003). In conclusion, hyperoxia was demonstrated to induce myopic shift both in neonatal and adult mice, which was attributed to the change in the CR rather than the AL. Elucidation of the mechanisms of hyperoxia and the application of this result to humans should be carried out in future studies.

## 1. Introduction

Due to the progress of neonatal care and the advancements in obstetrics, the rate of survival of premature babies has risen, which has resulted in an increase in the number of cases of retinopathy of prematurity (ROP). ROP presents a risk of severe visual impairment. The complications of ROP are known to be strabismus, glaucoma, retinal detachment, and myopia, in addition to proliferative changes [1,2,3,4].

ROP has been reported as a risk of myopia and its aggravated disorder, high myopia [5,6,7,8,9]. Quinn et al. first reported in 1992 that myopia is observed in ROP eyes [9], and then, Q’Connor et al., as well as Quinn et al., demonstrated the same results in their large-scale studies [7,10,11]. The prevalence of myopia varied among previous studies, ranging from 52.2% to 77% [12,13,14]. According to some authors [5,7,9], high myopia is more common in children with ROP than in children without it [14]. The study by Quinn et al. revealed that the rate of high myopia was 34.0% at 2 years of age and 37.9% at 3 years of age [7]. The prevalence and degree of myopia is well known to be correlated with a lower birth weight, prematurity, and the severity of ROP, but the mechanisms of myopia in ROP patients are not clearly understood [5,9].

Oxygen administration is one of the most important risk factors for ROP [15,16,17]. Oxygen-induced retinopathy (OIR) is a well characterized and the most used animal model mimicking ROP which is induced by an exposure to highly concentrated oxygen (hyperoxia) in the neonatal period. OIR rats were reported to show a myopic change [18]. In the meantime, hyperbaric oxygen therapy involves breathing pure oxygen in a pressurized room or through a tube. Hyperbaric oxygen therapy is known as a well-established treatment for decompression sickness, a hazard of scuba diving [19,20]. Myopia is reported to be induced by hyperbaric oxygenation therapy in adults [21,22].

Based on these facts above, our hypothesis is that myopia may be associated with excessive exposure to oxygen. A hyperoxia environment (exposed with an excessive oxygen supply) has not fully been demonstrated to affect myopia progression. However, there are some reports dealing with the effect of hyperoxia in myopia experiments and the definition of hyperoxia varies among studies; for example, 85% oxygen for 3 days or 75% oxygen for 5 days was administered as murine OIR models [23,24,25,26]. In this study, we determined 85% as hyperoxia and 21% as normoxia (normal oxygen condition) as the environment for the experiments. We investigated the change in refraction in the neonatal and the adult mouse model under the environment of highly concentrated oxygen.

The evaluation of myopic shift generally encompasses not only measuring refraction itself but also analyzing eye components such as the corneal curvature radius (CR), the central corneal thickness (CCT), the anterior chamber depth (ACD), and the lens thickness (LT) [27,28]. The measurement of lens elasticity is also considered an important part of the evaluation of myopia, to exclude the influence of hyperoxia on the lens such as cataract change. Most reports describe that the axial length (AL) is the biggest contributing factor to the myopic shift, but the associations between refractive errors and ocular components remain inconclusive [27]. There are some reports that show a correlation between refractive errors and the CR [29,30,31], whereas some studies reported higher ACD readings in myopia and lower ACD readings in hyperopia [29,32]. To know the associated factors with the refractive change induced by hyperoxia is undoubtedly significant to elucidate its mechanism, and thus, this study was intended to reveal which components are attributable to the phenomena observed in the eye in a hyperoxia environment.

## 2. Results

### 2.1. Neonatal Hyperoxia Model Mice Showed Myopic Shift

In the neonatal mice model, the hyperoxia group showed myopic shift, whereas the normoxia group, as a control, demonstrated a hyperopic trend (Figure 1a,b, Appendix A). The ALs of the hyperoxia group and that of the normoxia group were not significantly different both at Postnatal 30 days (P30d) and P47d (Figure 1c). The change in the AL from P30d to P47d was also not significant (Figure 1d). The change in the CR was not significant in the two groups (Figure 2a,b). The CCT was not significantly different between two groups at P30d. The CCT of the OIR group was larger than that of the control group at P47d (Figure 2c). The change in the CCT was not significantly different between the two groups (Figure 2d). The ACD was not significantly different between two groups at P30d and P47d (Figure 3a). The change in the ACD from P30d to P47d was significantly different in the hyperoxia group (Figure 3b). The LT of the hyperoxia group was smaller than that of the control group at P30d, and their difference was still significant at P47d. The degree of the change in the LT from P30d to P47d was not significant between the two groups (Figure 3c,d).

### 2.2. Hyperoxia Induced Myopia in Adult Mice

In adult mice, the refraction of the hyperoxia group shifted toward myopia one week after exposure to highly concentrated oxygen (Figure 4a,b, Appendix A). The AL and the change in the AL in both groups were not significantly different (Figure 4c,d). The CR of the hyperoxia group at P12w was significantly smaller than that of the control group (Figure 5a). The changes in CR in the two groups were significantly different 7 weeks after the baseline (7 w) (Figure 5b). The CCT in the hyperoxia group increased more than that in the normoxia group after high oxygen exposure (Figure 5c,d). In the hyperoxia group, the ACD increased one week after oxygen exposure and there was no difference between the two groups after that (Figure 6a,b). The LT showed the same trend as the ACD (Figure 6c,d). The elasticity of the lens at Postnatal 7 weeks (P7w) and P26w was not significantly different between the two groups (Figure 7a,b).

## 3. Discussion

To investigate the effect of hyperoxia on the eye, we measured the changes in ocular parameters such as refraction, AL, CR, and LT after exposing animals to highly concentrated oxygen, using neonatal and adult mice models. In this study, it was demonstrated that highly concentrated oxygen induced myopia both in neonates and adults. Interestingly, axial elongation, which usually happens in myopic eyes, was not observed in the current experiments.

A previous study showed that a rat OIR model induced myopia [33]. In the current study, we also observed a significant myopic shift in the mice of an OIR model, which corresponds to the neonatal hyperoxia group. Surprisingly, no significant difference in axial elongation was seen between the hyperoxia group and the normoxia group, which was consistent with that in the OIR rat model [33]. These phenomena match the fact that ROP children are often found to have myopia with normal axial length [5,34].

The degree of myopia in ROP was reported to be related to the depth of the anterior chamber, the thickness of the lens, and the change in AL, but not to keratometric values [1]. On the contrary, in other studies, myopia observed in ROP cases are shown to be due to corneal change, a decrease in the anterior chamber depth, lens thickening, short axial length, or abnormal oxygen metabolism [5,16,34]. Meanwhile, an increase in the ACD and an unchanged LT were observed in the hyperoxia group in this study. Although these results differed from the clinical ROP reports, an unchanged AL was consistent with them. It was difficult to determine which elements were the most contributory to the myopia progression; however, it was very interesting that AL elongation was not observed and the change in the CR was remarkable. The mechanism of myopic shift induced by hyperopia may be unique and should be investigated in the future studies.

Previous studies demonstrated that hyperbaric oxygenation can induce myopia in adult humans, and myopia induced by hyperbaric oxygenation therapy is mainly due to changes in the lens [21,22,35,36]. On the other hand, our study in adult mice showed that the thickness and the elasticity of the lens were not changed; nevertheless, myopia was induced by highly concentrated oxygen. In general, hyperbaric oxygen and highly concentrated oxygen are different from each other in nature, but to investigate the mechanism of myopic change caused by a high oxygen load, regardless of hyperbaric or highly concentrated oxygen, the comparison of these studies may be quite useful.

According to the results of this study, the refraction showed a transient hyperopic shift, and a larger CR and a smaller CCT were observed, at P19w in the adult hyperoxia group. These phenomena might be partially because some of the mice were suffering from corneal damage due to hyperoxia. The damage may have possibly healed spontaneously by P26w and the parameters were measured; however, we judged it was hard to repeat the measurement further, and then decided to abandon it. One of the possible explanation is that since the effect of hyperoxia on the cornea was more significant than anticipated, such damage occurred and myopia was induced because the CR did not enlarge as the mice grew.

Although the mechanism of the refractive change caused by an exposure to highly concentrated oxygen has not been clarified, some ideas exist regarding the mechanism, such as relative hypoxia in the retinopathy and dopamine oxygenation with reactive oxidative species (ROS). Since retinal vascularization supplies the sensory retina with a certain amount of oxygen pressure, any vascular alteration could modify this supply and produce transitory hypoxia conditions resulting in ischemia and reperfusion injury [37,38]. It has, however, been unclear how much a relatively low concentration of oxygen is related to the progression of myopia under hypoxic circumstances. It was also reported that ROS led to dopamine oxygenation [39]. One of the causes of myopia in OIR rats was a decrease of dopamine secretion [33]. Dopamine is a retinal neurotransmitter and is shown to be involved in the signaling cascade that controls eye growth [40]. Reduced retinal dopamine levels were observed in myopic eyes in animal models [40], and it can be suggested that myopia is caused by a decrease in active dopamine by its oxygenation under high-oxygen circumstances.

As for the possible molecular mechanism of myopia progression by hyperoxia, transcriptional factors induced by hypoxia may be considered. In addition to the most well-known hypoxia-related transcriptional factor—the hypoxia-inducible factor (HIF) [41]—a myopia suppressive transcriptional gene *Egr-1* is also upregulated by hypoxia [42]. Downregulation of *Egr-1* was observed during experimental myopia progression in various animal species [43,44,45,46,47]. Furthermore, *Egr-1* knock out mice showed spontaneous myopic shift of their refraction [46]. The *Egr-1* gene increased under hypoxia independently of HIF [42]. Thus, hyperoxia may induce downregulation of *Egr-1*, resulting in myopia progression. In fact, violet light, which has the shortest wavelength of visible light, was demonstrated to induce *Egr-1* and suppress myopia [48]. Dietary crocetin, one of the antioxidant carotenoids, was also revealed to suppress myopia through *Egr-1* activation [49]. These factors may be possible molecular candidates for the mechanism of hyperoxia on myopia to be elucidated in future studies.

The current results suggest the possibility that the progression of myopia in ROP eyes is due to an exposure to highly concentrated oxygen. This also may be applied to the myopic change in adults after exposure to highly concentrated oxygen. Although the mechanisms of the onset and the progression of myopia were found to be variable and complicated, it could be deduced that unnecessary exposure to highly concentrated oxygen should be avoided. Further investigation is warranted to elucidate the mechanism and the virtual correlation of oxygen exposure to myopic progression, and its application to human beings and establishment of other experimental models should be considered in future research. Molecular biological approaches in the investigation of myopic progression in the neonatal, ROP, and adult eyes exposed to highly concentrated oxygen could be the next step to elucidate the actual effect of oxygen on the eyes.

## 4. Materials and Methods

### 4.1. Animals

All procedures were performed in accordance with the National Institute of Health (NIH) guidelines for work with laboratory animals and the ARVO Animal Statement for the Use of Animals in Ophthalmic and Vision Research, and were approved by the Institutional Animal Care and Use Committee at Keio University (Approval number: 16017-(1); approval date: 25 October, 2017). Our study was also in compliance with ARRIVE guidelines. C57BL6/J mice (CLEA Japan, Japan) were raised in standard transparent mouse cages (29 × 18 × 13 cm) in an air-conditioned room maintained at 23 ± 3 °C under a 12-h dark/light cycle with free access to a standard diet (CE-2, CLEA Japan, Tokyo, Japan) and tap water. Four or five mice were kept in one cage.

### 4.2. Experimental Design

#### 4.2.1. Exposure to Highly Concentrated Oxygen

For the OIR model grouped as the neonatal hyperoxia group, P8d (8-day-old) C57BL6J mice (*n* = 6) were exposed to 85% oxygen for 3 days and then returned to room air conditions. Mice born on the same day and placed in a normal oxygen environment were used as a control (*n* = 6) grouped as the neonatal normoxia group. At P30d and P47d, ocular components including refraction, CR, AL, CCT, ACD, and LT were measured as mentioned below.

To evaluate the effect of hyperoxia on adult ocular components, 6-week-old (P6w) C57BL6J mice were exposed to an 85% concentration of oxygen (adult hyperoxia group, *n* = 6) or placed in normal air conditions (adult normoxia group, *n* = 6) for 3 days. Every component was measured before oxygen exposure as a baseline (P5w), and at P7w, P12w, P19w and P26w.

#### 4.2.2. Ocular Components Measurement

The efraction and the ocular components were measured using an infrared photorefractor (Steinbeis Transfer Center, Tübingen, Germany) and an SD-OCT system (Envisu R4310, Leica, Wetzlar, Germany). All measurements were performed under the condition of mydriasis by 0.5% tropicamide and 0.5% phenylephrine combination eye drops (Santen, Osaka, Japan), and general anesthesia with the combination of midazolam (Sandoz K.K., Tokyo, Japan), medetomidine (Domitor^®^, Orion Corporation, Espoo, Finland) and butorphanol tartrate (Meiji Seika Pharma Co., Ltd., Tokyo, Japan) (MMB). A 0.01 mL/g dose of MMB was administered intraperitoneally. The refractive error values were averaged over 100 measurements. The AL was determined from the anterior corneal surface to the retinal pigment epithelium along the corneal vertex reflection. The CCT was the distance from the anterior corneal surface to the posterior corneal surface, the ACD was from the posterior corneal surface to the anterior lens surface, and the LT was from the anterior lens surface to the posterior lens surface. The CCR was measured by an infrared keratometer (Steinbeis Transfer Center, Tübingen, Germany) [50]. The lens elasticity was measured by Softmeasure (Horiuchi Electronics Co. LTD., Tokyo, Japan) after enucleation under an overdose of anesthesia.

#### 4.2.3. Measurement of Lens Elasticity

The adult hyperoxia group (*n* = 4) and the adult normoxia group (*n* = 4) were used to examine the elasticity of the lenses. Enucleation was performed on 7-week-old mice, one week after the exposure to oxygen, and on 26-week-old mice, 20 weeks after the exposure to oxygen, under over-dosed general anesthesia to extract the lenses. The lens elasticity was determined by measuring the Young’s modulus, which is the modulus of longitudinal elasticity, using Softmeasure models of HG series (MSES) (Horiuchi Electronics Co. LTD., Tokyo, Japan). The device was installed under the condition of Force max 0.05 N, Speed 0.1 mm/s, initial head position −6.0 mm, spherical specimen diameter 1.5 mm, and Poisson’s ratio 0.45.

### 4.3. Statistical Analyses

The statistical analysis was performed as follows: after analyzing whether it showed a parametric or non-parametric distribution by the Shapiro–Wilk test, a *t*-test was applied to two group comparisons if it showed a normal distribution. For comparisons among three groups or more, one-way analysis of variance (ANOVA) with Post Hoc tests were used. The change in values from the baseline was determined as the subtraction of the baseline from the actual measured value. In order to eliminate the biases of eye growth, and the times of measurement, anesthesia, and eye drops, the comparison of the actual measured values or the changed values was performed at the same postnatal days and weeks both in the normoxia and hyperoxia groups. All results are expressed as mean ± standard deviation (SD). The results with *p*-values <0.05 were considered significant.

## Figures and Tables

**Figure 1 ijms-20-06014-f001:**
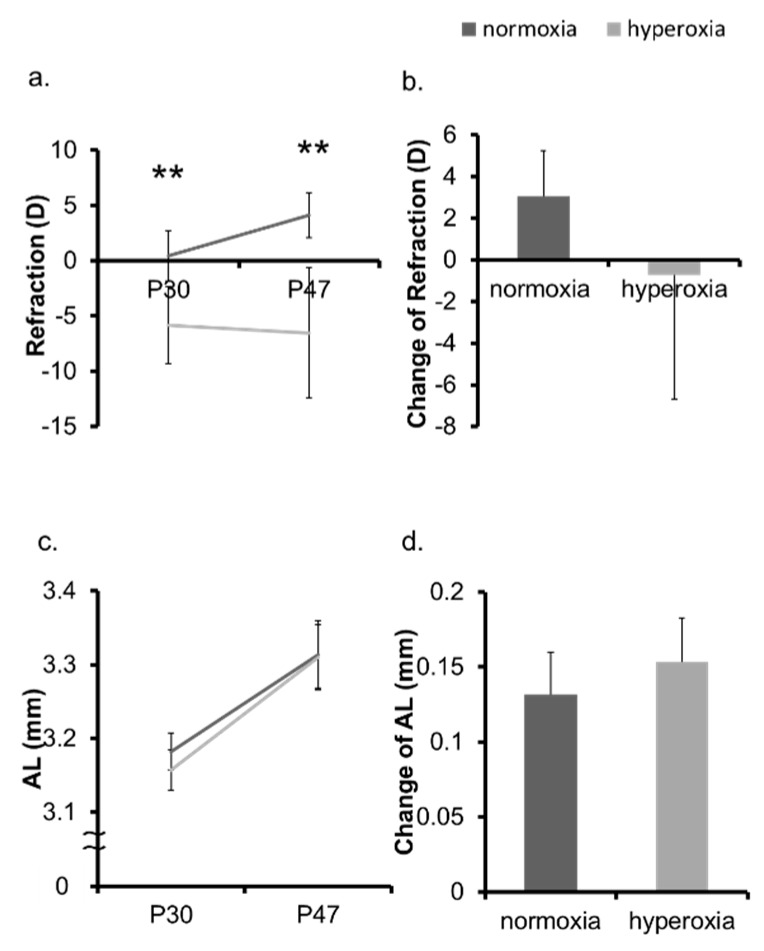
Change in refraction and AL in neonatal mice. The refraction and the AL of the normoxia group and the hyperoxia group were measured at P30d and P47d, respectively. (**a**) The refraction of the hyperoxia group shifted toward myopia compared to the normoxia group. (**b**) The change in the refraction of the hyperoxia group were reduced. (**c**) The AL of the hyperoxia group at P30d was not significantly different between the two groups. (**d**) The change in the AL was not significantly different between the two groups. ** *p* < 0.01, the bars represent mean +/− standard deviations. AL: axial length, P30d: 30 days old, P47d: 47 days old, D: diopter.

**Figure 2 ijms-20-06014-f002:**
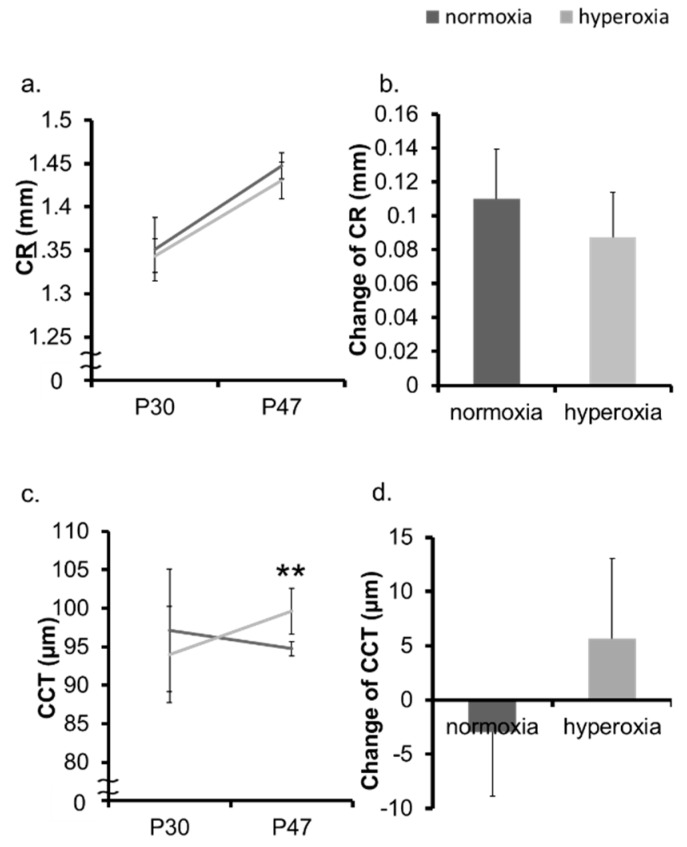
Change in CR and CCT in neonatal mice. The CR and the CCT of the normoxia group and the hyperoxia group were measured at P30d and P47d, respectively. The absolute value (**a**) and the change (**b**) in the CR of the OIR were not significantly different between the two groups. (**c**) The CCT in the normoxia group became thin, while that in the hyperoxia group became thick. (**d**) The change in the CCT was not significantly different between the two groups. ** *p* < 0.01, the bars represent mean +/− standard deviations. CR: corneal curvature radius, CCT: central corneal thickness, P30d: 30 days old, P47d: 47 days old.

**Figure 3 ijms-20-06014-f003:**
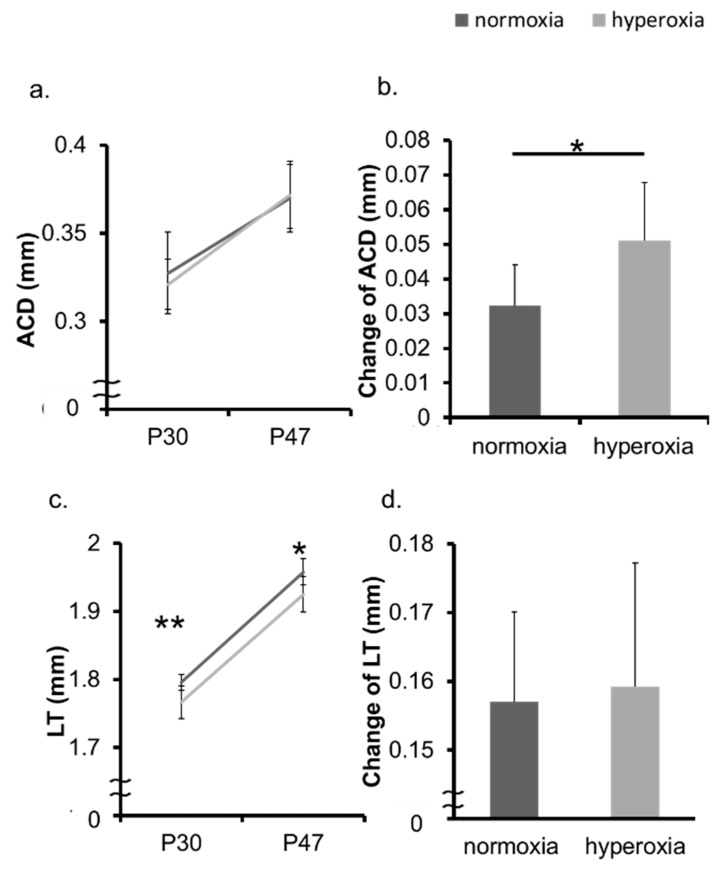
Change in ACD and LT in neonatal mice. The ACD and the LT of the normoxia group and the hyperoxia group were measured at P30d and P47d, respectively. (**a**) There was no significant difference in the absolute value of the ACD. (**b**) The change in the ACD was significantly different between the two groups. (**c**) The LT of the hyperoxia group was smaller than that of the normoxia group at P30d and P47d. (**d**) The degree of the change in the LT from P30d to P47d was not significant between the two groups. * *p* < 0.05, ** *p* < 0.01, the bars represent mean +/− standard deviations. ACD: anterior chamber depth, LT: lens thickness, P30d: 30 days old, P47d: 47 days old.

**Figure 4 ijms-20-06014-f004:**
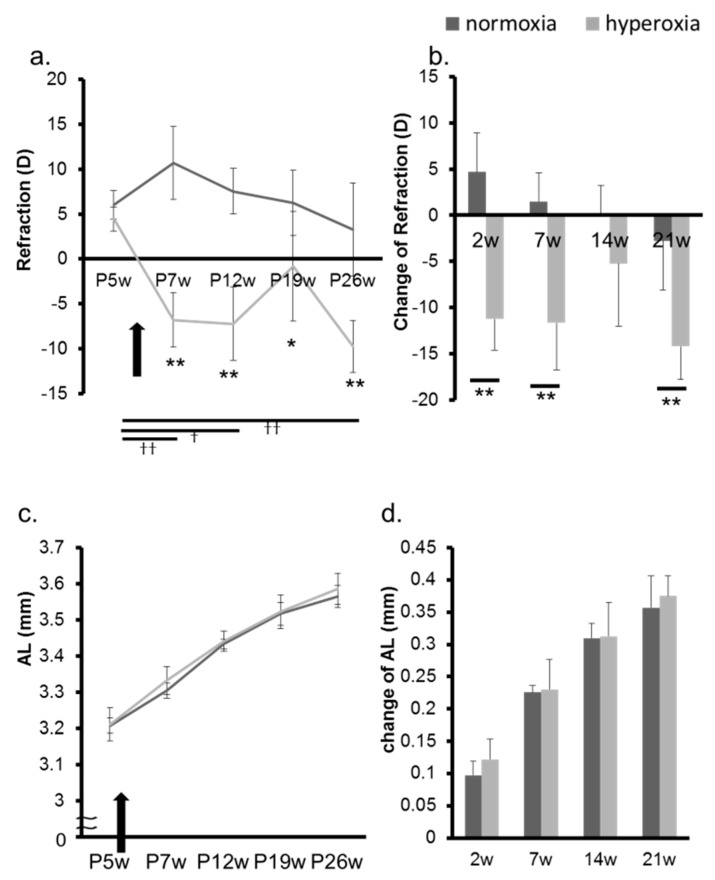
Change in refraction and AL in adult mice. In adult mice, the refraction and AL in the normoxia group and in the hyperoxia group were evaluated from P5w to P26w. A black arrow indicates the timing of the oxygen exposure. (**a**) The refraction of the hyperoxia group shifted toward myopia one week after the exposure to highly concentrated oxygen. (**b**) The changes in the refraction from the baseline at P5w are shown. The refraction of the hyperoxia group shifted toward myopia one week after being removed from the high-oxygen environment. The AL (**c**) and the change in the AL (**d**) in both groups were not significantly different. * *p* < 0.05, ** *p* < 0.01 (*t*-test, the normoxia group vs the hyperoxia group at each postnatal week) ^†^
*p* < 0.05, ^††^
*p* < 0.01 (ANOVA with Post Hoc test, the base line vs each postnatal week), the bars represent mean +/− standard deviations. AL: axial length, P5w: 5 weeks old, P7w: 7 weeks old, P12w: 12 weeks old, P19w: 19 weeks old, P26w: 26 weeks old, 2 w: 2 weeks from baseline at P5w, 7 w: 7 weeks from baseline at P5w, 14 w: 14 weeks from baseline at P5w, 21 w: 21 weeks from baseline at P5w, D: diopter.

**Figure 5 ijms-20-06014-f005:**
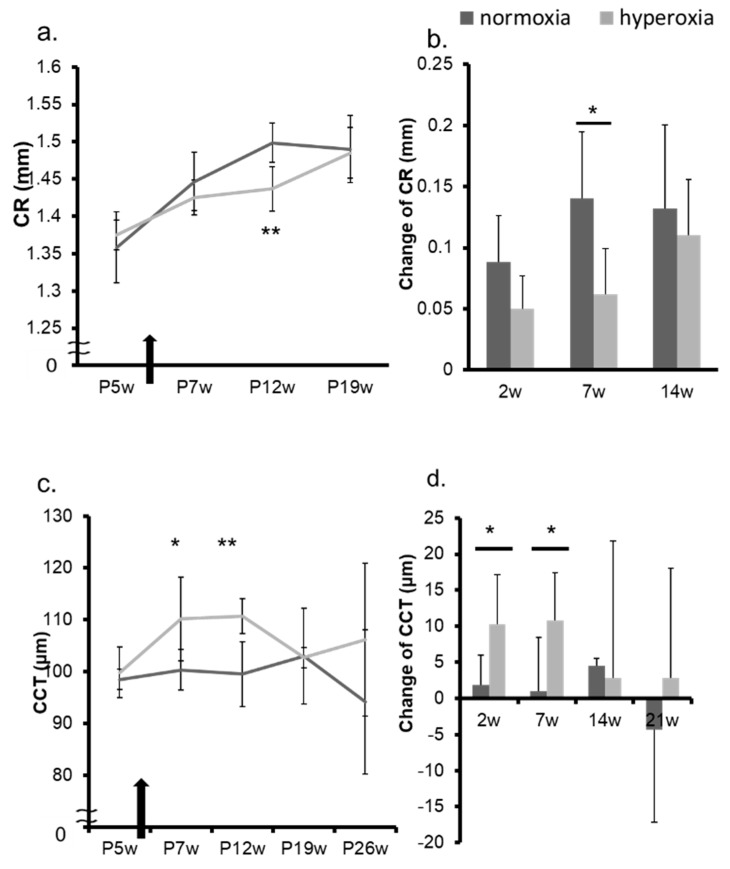
Change in CR and CCT in adult mice. In adult mice, the CR and CCT in the normoxia group and in the hyperoxia group were evaluated from P5w to P26w. A black arrow indicates the timing of oxygen exposure. (**a**) The CR of the hyperoxia group at P12w was significantly smaller than that of the normoxia group. (**b**) The changes in CR in the two groups were significantly different 7 weeks after baseline. The CR in the hyperoxia group increased less than that in the normoxia group after the high oxygen exposure. The CCT (**c**) and the change in the CCT (**d**) in the hyperoxia group increased more than that in the normoxia group at 2 and 7 weeks after the high oxygen exposure. * *p* < 0.05, ** *p* < 0.01, the bars represent mean +/− standard deviations. CR: corneal curvature radius, CCT: central corneal thickness, P5w: 5 weeks old, P7w: 7 weeks old, P12w: 12 weeks old, P19w: 19 weeks old, P26w: 26 weeks old, 2 w: 2 weeks from baseline at P5w, 7 w: 7 weeks from baseline at P5w, 14 w: 14 weeks from baseline at P5w, 21 w: 21 weeks from baseline at P5w.

**Figure 6 ijms-20-06014-f006:**
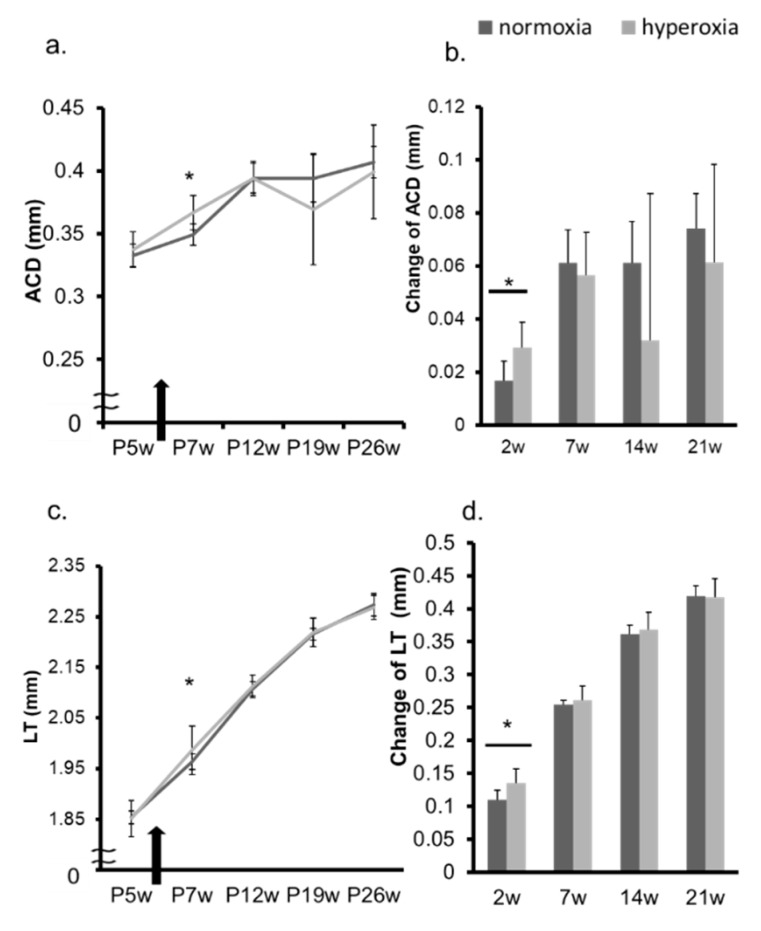
Change in ACD and LT in adult mice. In adult mice, the ACD and LT in the normoxia group and in the hyperoxia group were evaluated from P5w to P26w. A black arrow indicates the timing of oxygen exposure. In the hyperoxia group, the ACD (**a**) and the change in the ACD (**b**) increased one week after oxygen exposure and there was no difference between the two groups after that. The LT (**c**) and the change in the LT (**d**) showed the same trend as the ACD. * *p* < 0.05, the bars represent mean +/− standard deviations. ACD: anterior chamber depth, LT: lens thickness, P5w: 5 weeks old, P7w: 7 weeks old, P12w: 12 weeks old, P19w: 19 weeks old, P26w: 26 weeks old, 2 w: 2 weeks from baseline at P5w, 7 w: 7 weeks from baseline at P5w, 14 w: 14 weeks from baseline at P5w, 21 w: 21 weeks from baseline at P5w.

**Figure 7 ijms-20-06014-f007:**
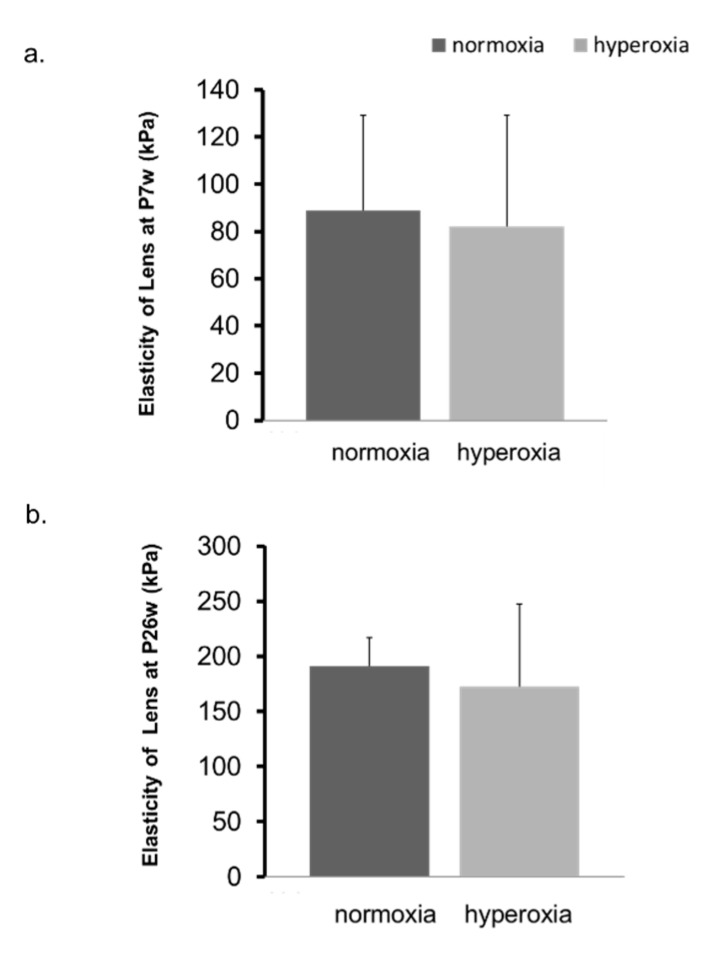
Change in elasticity of lenses in adult mice. The elasticity of lenses in the normoxia group and the hyperoxia group was examined at P7w (**a**) and P26w (**b**). There was no significant difference between the two groups. The bars represent mean +/− standard deviations. kPa: kilopascal, P7w: 7 weeks old, P26w: 26 weeks old.

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
