# Peer review of "Effects of Hyperoxia on the Refraction in Murine Neonatal and Adult Models"

_ijms, 2019, doi:10.3390/ijms20236014_

Round 1
Reviewer 1 Report
General
The study approaches the way hyperoxia affects parameters of the eyes of mice, including corneal curvature radius, axial length, lens elasticity and thickness. Producing of myopia (and further on high myopia) is thus investigated. The topic is of interest, the paper is in general well written, therefore it can be considered for publication in the journal, with a few corrections, as detailed bellow.
Specific
1) Please include in the Abstract all investigated parameters of mice eyes and point out all obtained conclusions.
2) Please explain all notations the first time they appear in the text, starting with ‘The P6w’ in the Abstract-and not just latter on in the text.
3) The authors should maybe explain for the readers terms like ‘normoxia’ and ‘hyperoxia’ in the Intro.
4) Please correct the Conclusions section, it now includes the template text.
Author Response
Point by Point Comments and Responses
Reviewer #1
General
The study approaches the way hyperoxia affects parameters of the eyes of mice, including corneal curvature radius, axial length, lens elasticity and thickness. Producing of myopia (and further on high myopia) is thus investigated. The topic is of interest, the paper is in general well written, therefore it can be considered for publication in the journal, with a few corrections, as detailed bellow.
Response
We appreciate your kind review of our manuscript giving us encouraging comments and valuable suggestions. It is so far practically the first study investigating the effects of hyperoxia on refractive changes in an animal model. We hope it will be of much help for researchers in the world to explore the mechanisms and preventive measures of myopia progression and to apply them to the clinical medicine in the future.
Specific
Please include in the Abstract all investigated parameters of mice eyes and point out all obtained conclusions.Response
Thank you for your suggestion and we revised our manuscript accordingly. We added all investigated parameters of mice eyes as below. In addition, we decided to delete the sentence regarding the lens elasticity because it was considered insignificant to elicit the conclusions of this study.
Line 21: added; (P47d -6.56±5.89 D, +4.11±2.02 D, p<0.001)
Line 22: added; (P47d, 3.31±0.04 mm, 3.31±0.05 mm, p=0.852)
Line 23: added; (P12w, -7.20±4.09 D, +7.52±2.54 D, p<0.001)
Line 24: added; (P12w, 3.44±0.03 mm, 3.43±0.01 mm, p=0.545)
Line 25: added; (P12w, 1.44±0.03 mm, 1.50±0.03 mm, p=0.003)
Line 26: deleted; The elasticity of the lens was not significantly different between the two adult groups.
Please explain all notations the first time they appear in the text, starting with ‘The P6w’ in the Abstract-and not just latter on in the text.
Response
We greatly appreciate your suggestion regarding the abbreviations. We carefully looked over the entire manuscript not to miss the explanation of the notations. We revised the manuscript so that all the abbreviations are explained when they appear the first time, as follows.
Line 16: Postnatal 6 weeks (P6w)
Line 67: corneal curvature radius (CR), central corneal thickness (CCT), anterior chamber depth (ACD), and lens thickness (LT). …axial length (AL)
Line 84: Postnatal 30 days (P30d)
Line 127: Postnatal 7 weeks (P7w)
The authors should maybe explain for the readers terms like ‘normoxia’ and ‘hyperoxia’ in the Intro.
Response
Thank you for your precious suggestions for us to explain to the readers terminology like normoxia and hyperoxia in the introduction section. We revised the manuscript as follows.
Line 58: added; Hyperoxia environment (exposed with an excessive oxygen supply) has not fully been demonstrated to affect myopia progression. However, there are some reports dealing with effect of hyperoxia in myopia experiments and the definition of hyperoxia varies among studies; for example, 85% oxygen for 3 days or 75% oxygen for 5 days was administered as murine OIR models [23-26]. In this study, we determined 85% as hyperoxia and 21% as normoxia (normal oxygen condition) environment for the experiments.
Please correct the Conclusions section, it now includes the template text.
Response
Thank you pointing out the very important issue. It was our mistake to leave the original instruction in the format provided by the journal. We intended to omit the conclusion section as the journal indicates that this section is not mandatory and would like to delete the section title of 5. Conclusions. Thank you.
Reviewer 2 Report
Manuscript Overview: Mori K et al. tested the hypothesis that myopia is associated with hyperoxia in neonatal and adult mice. Neonatal mice at postnatal day (P) 8 were exposed to 85% oxygen for 3 days then returned to room air (RA). Ocular components were assessed at P30 and P47. Adult mice were similarly exposed to oxygen for 3 days and ocular components determined before exposure (baseline), and at 5, 7, 12, 19, and 26 weeks post exposure. Age-matched controls were raised in RA and similar ocular components determined. The authors found that neonatal mice exposed to hyperoxia developed myopia at P30 and P47, concurrent with increased central corneal thickness (CCT) at P47, change in anterior chamber depth from P30-P47, and reduced lens thickness at P30 and P47. In adults, mice exposed to hyperoxia developed myopia at week 7, one week post hyperoxia with reductions in corneal curvature radius (CR) at 12 weeks post, and elevations in CCT at 7 and 12 weeks post. The authors conclude that hyperoxia induces myopic shift by altering CR.
Comments to the authors:
Abstract:
Spell out abbreviations at first mention e.g. (P8d; P7w, etc.). Correct grammar on lines 19 and 21 (change “significant larger” to “significantly larger”). The conclusions are stated as a rehashing of the results. Revise the statement to provide the meaning of the findings and give a take-home message.Introduction:
State the objectives of the study. Provide further information regarding the usefulness of the ocular measurements in myopia to orient the reader.Results:
Line 60, spell out CR at first mention. Some of the error bars show very large variations, likely due to small sample sizes of n=6). This indicates that some of the data did not follow Gaussian distribution. In this case, Student’s t-test is not appropriate. How were the non-normally distributed data analyzed? Figure 4a shows comparisons from baseline (per legend). This means that 4 groups were compared. Based on the statistical analyses, t-test is not appropriate. Revise the statistical analyses to include appropriate statistics for >2 groups. Consultation with a statistician is advised since there seems to be non-normal data based on the large standard deviation.Discussion:
The discussion could be largely improved with emphasis and further in-depth analysis of the findings in the present study.Methods:
Describe how the change in values from baseline were determined. Provide a rationale for studying ocular components so frequently in the adult mice, particularly due to combined influence of anesthesia, midazolam/butorphanol tartrate, and eye drops. Revise statistical analyses to include how non-normally distributed data were analyzed and how >2 groups were analyzed.Author Response
Please see the attachment.

Round 2
Reviewer 2 Report
The authors have satisfactorily addressed all comments and queries raised, and the manuscript has been improved.